# A multilevel analysis of short birth interval and its determinants among reproductive age women in developing regions of Ethiopia

**Setognal Birara Aychiluhm**[1]*, **Abay Woday Tadesse**[1], **Kusse Urmale Mare**[2], **Mohammed Abdu**[3], **Abdusemed Ketema**[3]

**1** Department of Public Health, College of Medicine and Health Sciences, Samara University, Samara, Ethiopia, **2** Department of Nursing, College of Medicine and Health Sciences, Samara University, Samara, Ethiopia, **3** Department of Midwifery, College of Medicine and Health Sciences, Samara University, Samara, Ethiopia

* geez4214@gmail.com

**Data Availability Statement:** Data underlying this study can be found at https://dhsprogram.com/data/dataset/Ethiopia_Standard-DHS_2016.cfm.

## Abstract

### Background

Short Birth Interval negatively affects the health of both mothers and children in developing nations, like, Ethiopia. However, studies conducted to date in Ethiopia upon short birth interval were inconclusive and they did not show the extent and determinants of short birth interval in developing (Afar, Somali, Gambella, and Benishangul-Gumuz) regions of the country. Thus, this study was intended to assess the short birth interval and its determinants in the four developing regions of the country.

### Methods

Data were retrieved from the Demographic and Health Survey program official database website (http://dhsprogram.com). A sample of 2683 women of childbearing age group (15–49) who had at least two alive consecutive children in the four developing regions of Ethiopia was included in this study. A multilevel multivariable logistic regression model was fitted to identify the independent predictors of short birth interval and Akaike's Information Criterion (AIC) was used during the model selection procedure.

### Results

In this study, the prevalence of short birth interval was 46% [95% CI; 43.7%, 47.9%]. The multilevel multivariable logistic regression model showed women living in rural area [AOR = 1.52, CI: 1.12, 2.05], women attended secondary education and above level [AOR = 0.27, CI: 0.05, 0.54], have no media exposure [AOR = 1.35, CI: 1.18, 1.56], female sex of the index child [AOR = 1.13, CI:1.07,1.20], breastfeeding duration [AOR = 0.79, CI: 0.77, 0.82], having six and more ideal number of children [AOR = 1.14, CI: 1.09, 1.20] and having preferred waiting time to birth two years and above [AOR = 0.86, CI: 0.78, 0.95] were the predictors of short birth interval.

**Funding:** The authors received no specific funding for this work.

**Competing interests:** The authors have declared that no competing interests exist.

## Conclusions

The prevalence of short birth intervals in the developing regions of Ethiopia is still high. Therefore, the government of Ethiopia should work on the access of family planning and education in rural parts of the developing regions where more than 90% of the population in these regions is pastoral.

## Introduction

The World Health Organization (WHO) and Ethiopian Demographic and Health Survey (EDHS) reports on birth spacing recommended a birth to conception interval of at least 24 months in two consecutive births [1, 2].

Demographic and Health Survey (DHS) data from 18 developing countries (Africa, Asia, Latin America, and the Middle East) and an International comparison study of 77 countries using DHS data revealed that a birth interval of three or more years interval improves the survival status of mothers, under-five children and infants [3, 4].

Ethiopia is the second-most populous country in Africa, with a population size of more than 100 million and a fertility rate of 4.6 children per woman [2, 5]. Like many other African countries, Ethiopia has shown so far little change in fertility reduction because of socio-cultural and religious factors [6]. For instance, first marriage at an early age, desire for more children, and low contraceptive utilization related to religious issues influence the status of fertility[7, 8].

In developing nations, more than 200 million women either want to space or limit pregnancies and yet they lack access to modern family planning options [1, 9–12]. Demographic Health Survey (DHS) studies revealed a high level of Short birth intervals (SBIs) in the region (Rwanda: 20%, Uganda: 25.3%, and Cameroon: 21.3%) [13]. In Ethiopia, the prevalence of SBI (i.e. Birth interval < 24 months) ranges between 23.3% and 59.9% [14–17].

Globally, a birth interval of fewer than 18 months is associated with increased risk for neonatal mortality, infant mortality, under-five mortality, and maternal mortality [4, 6, 9, 15–21]. Similarly, Ethiopia has experienced a significant number of neonatal mortality and infant associated with short birth interval compared to the overall average rate of infant and neonatal mortality reported in Africa [18].

Studies conducted across the globe have identified various factors associated with SBI. These include; maternal age, maternal education level, husband education level, death of the index child, sex preference of the parents, no use of contraceptives, the ideal number of children, socio-cultural factors, religion, short breastfeeding duration (less than 24 months), and poor wealth index [14, 17, 22–27].

The Sustainable Development Goals (SDGs) of 2030, which combine multisystem strategies at global, regional, and national levels, have three focuses to ensure healthy lives and promote wellbeing for all at all ages. Of these goals, one main objective is to reduce the neonatal mortality rate to lower than 12 per 1,000 live births [28–30] which is at a steady stage in developing nations. According to the 2019 mini EDHS, infant mortality rate was 43 deaths per 1,000 live births and under-5 mortality rate was 55 deaths per 1,000 live births in Ethiopia [31]. By 2030, Ethiopia aiming to reduce neonatal mortality to at least as low as 12 per 1000 live births and under-five mortality to at least as low as 25 per1000 live births [32].

Despite the implementation of various strategies and interventions at global and national levels to decline burden of under-five children, infant and neonatal mortality rates [10, 33–36], short birth spacing remains one of the leading causes of child mortality [4, 37] in developing

nations [18, 21]. In Ethiopia, still, 22% of women have an unmet need for family planning(FP) with 35% of contraceptive discontinuation rates [2]. Thus, this may contribute to the high level of SBI in the country. Besides, studies conducted in Ethiopia were limited to developed regions and inconclusive to show the determinants of short birth interval in developing regions. Moreover, previous studies conducted [12, 17, 27, 38] in the country had not been identified the community level determinants of short birth interval. Furthermore, using a single-level logistic regression analysis technique to analyze data that has a hierarchical structure nature (that is women nested within communities) violates the independence assumptions of regression [39, 40]. Hence, to address these limitations, and to further estimate the significant effect of individual and community-level factors in developing regions of Ethiopia, this study used multilevel logistic regression analysis.

The results of this study will offer crucial information for policymakers, program planners, and other stakeholders to plan and implement proper interventions to prevent short birth interval in developing regions of the country.

Therefore, this study was aimed to address both the individual and community-level determinants of short birth interval among women resided in the four developing regions (Afar, Somali, Benshandul-Gumz, and Gambella) of Ethiopia.

## Methods and materials

### Study area and data source

The study was conducted in developing regions of Ethiopia which are found mainly in lowland parts of the country. These regions are; Afar, Somali, Gambella, and Benishangul-Gumuz regions. These four regions are not well achieving most of the indicators related to health, human development and Millennium Development Goals compared to other developed regions of Ethiopia [41]. The main lifestyle of these regions depends on animal livestock and farming. Hence, the communities resided in these regions are nomadic ethnic groups and highly mobile which are not suited to the existing health system of the country [42–44]. Besides, in developing regions of the country, women in reproductive-age group are inaccessible to modern contraceptives, more over in these developing regions of the country, there are socio-cultural and religious barriers towards the utilization of birth control methods [43, 45, 46].

The data were retrieved from the Demographic and Health Survey (DHS) program official database website (http://dhsprogram.com), that was conducted in nine regions (Tigray, Afar, Amhara, Oromia, Somali, Benishangul-Gumuz, Southern Nations Nationalities and Peoples Region (SNNPR), Gambella, and Harari), and two city administrations (Addis Ababa and Dire-Dawa) of Ethiopia from January 18, 2016, to June 27, 2016.

To conduct the 2016 Ethiopian Demographic and Health Survey (EDHS), a two-stage stratified cluster sampling technique has been employed. Enumeration areas were selected in the first stage. In the second stage, 28 households per enumeration area were selected with an equal probability of systematic selection per Enumeration Area (EA). Nationally, a total of 645 EAs were selected with probability proportional to EA size, and nationally a total sample size of 16,515 women aged 15–49 years was collected.

The study populations for this study were 2683 (388 from Afar, 1706 from Somali, 489 from Benishangul-Gumuz, and 100 from Gambella) women who had at least two consecutive live births in the 5 years preceding the survey, nested within four developing regions of the country [2]. For this study, all women of childbearing age group (15–49) having at least two alive children in four developing regional state governments of Ethiopia were included for our analysis.

Women who had never been married and those who have multiple births out of wedlock were excluded from the analysis.

## Study variables

**Dependent variable.** Short Birth Interval. The outcome variable of this study was a short birth interval (SBI) which was dichotomized into "Yes = 1/ No = 0" form. A birth that occurred at less than 24 months following a previous birth in two successive births was classified as having SBI, according to WHO recommendation [1]. The birth interval was calculated as the time that elapsed between the birth date of the first child and the birth date of the second child [47].

**Independent variables.** All the independent variables were selected based on reviewed different literature [9, 12, 13, 15, 17, 23, 27, 37] and those independent variables were classified into individual-level variables and community-level variables. Individual-level variables were sex of index child, age at marriage, mother's age at first birth, parity (number of live births), women's education level, husband's education level, husband's occupation, wealth index, respondents occupation, religion, exposure to any mass media, survival status of the index child, ideal number of children, preferred waiting time to birth, number of living children, duration of breastfeeding (in months), and contraceptive utilization. Community-level were region, type of residence, and cluster.

## Media exposure

To measure exposure to media in the 2016 EDHS, watching television (TV), listening to radio, and reading newspaper at least once a week were considered. The tree media channels have categories "all at least once a week", "both at least once a week" and "no accesses at least once a week". Therefore, a new variable called media exposure was generated by combining the three media sources (TV, Radio, and Newspaper), Then media exposure was labeled as "Yes" if respondents had exposure at least one of the media channel and labeled "No" if respondents did not have any exposure to either of the three media channels.

## Multi-collinearity

The presence of multicollinearity among independent was checked using Variance Inflation Factor (VIF) taking cut off value of 10. Variables having a VIF value of less than 10 were considered as the absence of multicollinearity.

## Data analysis

**Descriptive statistics.** Based on the recommendation of EDHS, proportions and frequencies were estimated after applying sample weights to the data to adjust for disproportionate sampling and non-responses. Since the allocation of the sample in the EDHS to different regions as well as urban and rural areas were non-proportional. A detailed clarification of the weighting process can be found in the 2016 EDHS report [48]. Categorization was done for continuous variables using information obtained from different literatures, and re-categorization was done for categorical variables accordingly to make suitable for analysis. The analysis was performed using Stata version 15.0.

**Bivariable multilevel analysis.** The effect of each independent variable (both individual and community-level) on the dependent variable was checked at a p value of 0.25. Variables in which p-value of less than 0.25 in the bivariable multilevel logistic regression analysis were considered as candidates for multivariable multilevel logistic regression analysis.

**Multivariable multilevel analysis.** Due to the hierarchical nature of the 2016 EDHS data (i.e., mothers are nested within clusters), to account this clustering effect, a multivariable multilevel logistic regression analysis was applied to determine the effects of each predictor of SBI.

**Model building and comparison.** Four models containing variables of interest were fitted for this study.

**Model I (Empty model)** was fitted without explanatory variables to test random variability in the intercept and to estimate the intra-class correlation coefficient (ICC) and Proportion Change in Variance (PCV).

**Model II** assessed the effects of individual-level predictors,

**Model III** assessed the effects of community-level predictors and

**Model IV** (**Full model**) examined effects of both individual and community-level characteristics simultaneously.

Akaike's Information Criterion (AIC) was used to select the model and the model with low AIC value was considered as a best-fitted model. Based on AIC the full (model with individual and community-related variables) model has the smallest AIC value among the model considered, therefore the full model best fits the data. AOR with 95% Confidence interval in the multivariable model was used to select variables that have a statistically significant association with short birth interval.

**Ethical consideration.** The data were accessed from the Demographic and Health Survey (DHS) website (http://www.measuredhs.com) after getting registered and permission was obtained (AuthLetter_136950). The accessed data were used for this registered research only. The data were treated as confidential and no effort was made to identify any household or individual respondent.

## Results

### Descriptive statistics of the study variables

Out of the total respondents, 2,287 (84.2%) women were living in rural site, 2319 (86.4%) of the women were Muslim religion faith followers, 2281 (85.1%) were not attended formal education, 2141 (79.8%) of the women were agricultural workers, and 1695 (63.2%) of the respondents had poorest wealth index. In this study, the average breastfeeding duration for the preceding index child was 64 ± 0.03 standard deviation (SD) months. The study showed 2224 (82.9%) of the respondents did not have media exposure towards the short birth interval and 1651 (61.5%) of the women reported having more than six ideal numbers of children including the current birth. The study also revealed that 2508 (93.5%) of the women included in the study were not using any contraceptive methods (**Table 1**).

### Prevalence of short birth interval

Overall, 967(46% (95% CI; 43.7%, 47.9%)) of the women had experienced short birth interval, of these, 518 (53.6%) women had SBI which age at birth ≤ 18 years, 372(38.4%) ranges from 19–24 years and the rest 69(6.1%) were 25 years and above. Besides, from those women who had experienced short birth interval, the majority (707) of them were from Somali regional state.

### Determinants short birth interval

**Empty multilevel logistic regression model (Null model).** From the null model variance of the random factor was 0.21 with a 95% confidence interval of (0.05, 0.84), showing heterogeneous areas. Since the variance estimate, which is greater than zero, it indicates that there are

**Table 1. Weighted socio-demographic, reproductive, behavioral and child status-related characteristics of study participants, EDHS,2016 [N = 2683].**

| Variable | Category | Frequency | Percent |
|---|---|---|---|
| Residence | Urban | 396 | 14.8 |
| | Rural | 2,287 | 85.2 |
| Religion | Muslim | 2319 | 86.4 |
| | Others+ | 364 | 13.6 |
| Mothers age at first marriage | Less than 18 years | 1674 | 62.5 |
| | 18 and above | 1006 | 37.6 |
| Respondent's educational status | No education | 2281 | 85.0 |
| | Primary | 323 | 12.0 |
| | Secondary and above | 79 | 3.0 |
| Husband's educational status | No education | 1819 | 74.3 |
| | Primary | 382 | 15.6 |
| | Secondary and above | 247 | 10.1 |
| Respondent's occupation | Agriculture | 2141 | 79.8 |
| | Professional | 412 | 15.3 |
| | Others++ | 130 | 4.9 |
| Husband's occupation | Agriculture | 1622 | 66.1 |
| | Professional | 470 | 19.1 |
| | Others+++ | 357 | 14.6 |
| Wealth Index | Poorest | 1695 | 63.2 |
| | Poorer | 297 | 11.1 |
| | Middle | 191 | 7.1 |
| | Richer | 194 | 7.3 |
| | Richest | 305 | 11.4 |
| Sex of child | Male | 1377 | 51.3 |
| | Female | 1306 | 48.7 |
| Ideal number of Children | Less than 6 | 1032 | 38.5 |
| | 6+ | 1651 | 61.5 |
| Survival of index child | Yes | 2351 | 87.6 |
| | No | 332 | 12.4 |
| Preferred waiting time to birth | Less than 2 years | 1054 | 39.3 |
| | 2 and above years | 1629 | 60.7 |
| Use contraceptive | Yes | 175 | 6.5 |
| | No | 2508 | 93.5 |
| Media Exposure | Yes | 459 | 17.1 |
| | No | 2224 | 82.9 |

**Key:** Other+ = catholic, orthodox, protestant, other and traditional, Other++ = skilled manual, unskilled manual and other, Other+++ = Other EDHS category and laborer.

enumeration (cluster) area differences in short birth interval among women in four developing regional states in Ethiopia, and thus multilevel analysis should be considered as an appropriate approach for further analysis.

The intra-cluster correlation coefficient (ICC) which indicated that 6% of the total variability in short birth interval is due to differences across cluster areas, with the remaining unexplained 94% attributable to individual differences. The Proportion Change in Variance (PCV) indicated that 81% of the variation in short birth interval across communities was explained by both individual and community level factors included in the full model (**Table 2**).

**Table 2. Community-level variance of two-level mixed-effect logit models predicting short birth interval, EDHS 2016.**

| Random effect | Null model | Full model |
|---|---|---|
| Community-level variance | 0.21 | 0.04 |
| ICC (%) | 6 | 0.01 |
| PCV (%) | Reference | 0.81 |
| Model fitness statistics (AIC) | 2866 | 812 |

**Multilevel multivariable logistic regression model (Full model).** In the multilevel multi-variable logistic regression model, both the individual and community level factors were fitted simultaneously. Thus, residence site, women's educational status, media exposure, sex of the index child, breastfeeding duration, the ideal number of children and preferred waiting time to birth were statistically associated with a short birth interval at 95% confidence level.

After adjusting for covariates; the odds of the short birth interval among women in a rural area was 1.52 times higher compared to those living in an urban area (AOR = 1.52, CI: 1.12, 2.05).

This study revealed that while breastfeeding duration of the index child increase by one month, the odds of SBI among women decrease by 21% (AOR = 0.79, CI: 0.77, 0.82).

In this study, women having female sex of the index child had 1.13 times greater risk of short birth interval compared to those women having male index children (AOR = 1.13, CI:1.07,1.20).

Keeping other covariates constant, women who attended secondary education and above levels were 27% less likely to have SBI compared to women without formal education (AOR = 0.27, CI: 0.05, 0.54).

The odds of the short birth interval among women who did not have exposure to any media about short birth interval before or during the index child was 1.35 times greater compared to those women did have exposure to any media (AOR = 1.35, CI: 1.18, 1.56).

In this study, women who have preferred waiting time to birth two years and above were 14% less likely to have short birth interval compared to those who had preferred waiting time less than two years (AOR = 0.86, CI: 0.78, 0.95). Moreover, women who have a desire of six or more children had 1.14 times greater risk of short birth interval compared to those women having a desire of fewer than six children (AOR = 1.14, CI: 1.09, 1.20) (**Table 3**).

## Discussion

Women's physiological regression is the only hypothetical causal mechanism that has been proposed to explain the association between short birth spacing and maternal health and adverse birth outcomes [49]. This study aimed to determine the prevalence and determinants of short birth interval among women in developing regions of Ethiopia using the EDHS 2016 dataset. This study revealed that residence site, women's educational status, media exposure, sex of the index child, breastfeeding duration, ideal number of children, and preferred waiting time were the independent predictors of short birth interval in the four developing regions of Ethiopia.

In this study, the prevalence of short birth interval in the four developing regions of Ethiopia is 46% [95% CI; 43.7%, 47.9%]. This finding is higher than a study conducted in Northern Ethiopia (23.3%) [14], a study done in Jimma, Southwest Ethiopia (27%) [15], Dabat district, Northwest Ethiopia (39.1%) [23], Arsi Zone, Ethiopia (17.3%) [24], Northern Ethiopia (40.9%) [50], and rural Bangladesh (24.6%) [37]. This discrepancy could be due to the fact that the current study is carried out in the developing regions of the country, where women in the

**Table 3. Multilevel multivariable logistic regression of the individual and community-related variables associated with short birth interval, EDHS 2016.**

| Variables | Categories | Short Birth Interval Status | | AOR (95%CI) |
|---|---|---|---|---|
| | | Yes | No | |
| Residence site | Urban (ref) | 122 (12.6) | 171 (14.9) | 1.00 |
| | Rural | 845 (87.4) | 972 (85.1) | 1.52 (1.12, 2.05) * |
| Wealth index | Poorest (ref) | 653 (67.5) | 705 (61.7) | 1.00 |
| | Poorer | 96 (10.0) | 138 (12.1) | 0.94 (0.75, 1.17) |
| | Middle | 70 (7.2) | 77 (6.7) | 1.40 (0.80, 2.43) |
| | Richer | 58 (5.9) | 93 (8.2) | 1.06 (0.45, 2.47) |
| | Richest | 91 (9.4) | 129 (11.3) | 1.69 (0.87, 3.28) |
| Women education | No education (ref) | 870 (89.9) | 971 (85.0) | 1.00 |
| | Primary | 85 (8.8) | 138 (12.1) | 0.80 (0.63, 1.01) |
| | Secondary and above | 13 (1.3) | 33 (2.9) | 1.27(1.05, 1.54) * |
| Media exposure | No (ref) | 834 (86.4) | 935 (81.8) | 1.00 |
| | Yes | 132 (13.6) | 208 (18.2) | 0.74 (0.64, 0.85) * |
| Husband occupation | Agriculture (ref) | 600 (66.9) | 701 (67.8) | 1.00 |
| | Professional | 167 (18.6) | 191 (18.5) | 0.86 (0.65, 1.12) |
| | Others+ | 130 (14.5) | 142 (13.8) | 0.80 (0.64, 1.01) |
| Religion of respondent | Muslim | 890 (92.0) | 956 (83.7) | 1.65 (0.85, 3.20) |
| | Others++ | 78 (8.0) | 186(16.3) | 1.00 |
| Age at marriage | Less than 18 years | 588 (60.8) | 734 (64.3) | 1.05 (0.88, 1.24) |
| | 18 and above years | 379 (39.2) | 408 (35.7) | 1.00 |
| Age at birth of index child | ≤18 years | 518 (53.6) | 656 (57.5) | 0.90 (0.64, 1.29) |
| | 19–24 years | 372 (38.4) | 417 (36.5) | 0.94 (0.75, 1.18) |
| | 25 and above years | 78 (8.0) | 69 (6.1) | 1.00 |
| Sex of child | Male | 495 (51.2) | 584 (51.1) | 1.00 |
| | Female | 472 (48.8) | 559 (48.9) | 1.13 (1.07, 1.20) * |
| Number of live births | | | | 1.10(0.22, 1.18) |
| Number of living children | | | | 0.95(0.91, 1.00) |
| Ideal number of Children | Less than 6 (ref) | 329 (34.0) | 457 (40.0) | 1.00 |
| | 6+ | 638 (66.0) | 685 (60.0) | 1.14 (1.09, 1.20) * |
| Survival of index child | No (ref) | 155 (16.1) | 91 (8.0) | 1.00 |
| | Yes | 812 (83.9) | 1052 (92.0) | 0.52 (0.42, 1.62) |
| Preferred waiting time to birth | Less than 2 years (ref) | 414 (42.8) | 409 (35.8) | 1.00 |
| | 2 and above years | 554 (57.2) | 733 (64.2) | 0.86 (0.78, 0.95) * |
| Use contraceptive | No | 934 (96.6) | 1058 (92.6) | 1.36 (0.61, 3.04) |
| | Yes (ref) | 33 (3.4) | 84 (7.4) | 1.00 |
| Breastfeeding duration (months) | | | | 0.79 (0.77, 0.82) * |

ref = reference

* statistically significant variables at 95% confidence interval, Other+ = Other EDHS category, Other++ = catholic, other and traditional.

reproductive-age group are inaccessible to modern contraceptives. Besides, there are also socio-cultural barriers [51, 52] upon the utilization of birth controls in the developing community compared to the other parts of Ethiopia.

However, the prevalence is lower than a study conducted in Lemo district, Southern Ethiopia (57%) [12], Jimma Zone, Southwest Ethiopia (59.9%) [16], Tanzania (48.4%) [9], and Kassala, Eastern Sudan (60.6%) [53]. This could be explained by small sample size in the previous studies and the difference in study designs. In addition, this variation could be explained by a

difference in cut-off values used to determine SBI. Those previous studies considered SBI if birth interval less than 36 months, while our study defined it as less than 24 months.

In this study, the odds of short birth interval among women in reproductive-age group living in the rural area is 1.44 times higher compared to those women living in an urban area. This is similar to a study conducted in Lemo district, Southern Ethiopia [12], and a study done in the Democratic Republic of Congo [13]. This could be justified by women living in the rural sites are socio-economically disadvantaged [37] and inaccessible to modern contraceptive methods. Thus, they are more likely to experience a short birth interval compared to women residing in an urban area of the country.

This study revealed that as breastfeeding duration of the index child increase by one month, the odds of SBI among childbearing women decrease by 79%. This finding is similar to a study done in Serbo Town, Southwest Ethiopia [16], a systematic review of 58 observational studies [49], Northern Ethiopia [14], Dodota district, Southern Ethiopia [24], Northern Ethiopia [50], and Egypt [54]. Optimal breastfeeding prolongs the length of time between two consecutive births. women's physiological regression is the causal mechanism that has been proposed to explain the association between birth spacing and prolonged breastfeeding [49]. Consequently, the longer the duration of breastfeeding the women practicing for the index child, the lesser the risk of being short birth interval for the succeeding birth.

In this study, women having female sex of the index child had 1.13 times greater risk of short birth interval compared to those women having male index children. This finding is consistent with study done in Serbo Town, Southwest Ethiopia [16], rural developing communities of Southern Ethiopia [17], Arba Minch district, Ethiopia [27], and Northern Ethiopia [50]. In the developing community of Ethiopia, parents and their community members have male preference than female children. This sex preference is usually related to the families' interest in being safeguarded from enemies by their young male children. In addition, since the parents' lifestyle is related to livestock, they need more male children for the sake of keeping their cattle. Thus, the preceding index child being female has been contributed to the risk of a short birth interval to get more male children.

Women who attended secondary education and above levels were 0.27 less likely to have SBI compared to women without formal education. This finding is consistent with the study done in Tanzania (48.4%) [9], Democratic Republic of Congo [13], rural developing communities of Southern Ethiopia [17], Serbo Town, Southwest Ethiopia [16], Arba Minch district, Ethiopia [27], and Kassala, Eastern Sudan [53]. When the education status of the women increased, the knowledge and awareness of the women upon the consequences of short birth interval on maternal and child health will also be optimized. Thus, women attending secondary education level and above have a lower risk of short birth interval compared to women who have no education.

The odds of short birth interval among women who did not have exposure to any media about short birth interval before or during the index child was 1.35 times higher compared to those women who had exposure to any media. This finding is in line with the studies conducted in Bangladesh [55, 56]. Therefore, women who have information about short birth interval through any media channel are expected to have a better understanding of the negative impact of short birth interval on maternal and children's health. As a result, women who have no exposure for any media are more likely to experience short birth interval than those have any media exposure.

In this study, women with two years and above preferred waiting time to birth were 14% less likely to have short birth interval compared to those who had preferred waiting time less than two years. Moreover, women who have a desire to have six or more children had 1.14 times greater risk of short birth interval compared to those with a desire of fewer than six

children. In the developing community of Ethiopia, parents and their community members have a desire for more children because of socio-cultural and religious interests. For instance, the majority of the community in the four developing regions of Ethiopia are Muslim religious faith followers, in which the use of modern contraceptives for child spacing does not have been practiced yet [52]. This finding is also consistent with a study conducted in Northern Ethiopia [14], Jimma Zone, Southwest Ethiopia [15], rural developing communities of Southern Ethiopia [17] and Kassala, Eastern Sudan [53]. Furthermore, the lifestyle of the community in the developing regions is purely dependent on livestock, in which having more children is considered as advantageous to get more keeper for their cattle. Thus, this perception of the developing community towards more children has been one of the contributors to short birth interval in these regions of the country. In this study, there was no significant association between wealth index of the household and birth interval of women.

## Conclusion

The prevalence of short birth interval in the developing regions of Ethiopia is still optimally high. In the multilevel multivariable logistic regression model; residence site, women's educational status, media exposure, sex of the index child, breastfeeding duration, ideal number of children, and preferred waiting time were the independent predictors of short birth interval in the four developing regions of Ethiopia. Therefore, the government of Ethiopia should work on the access to family planning and education in rural parts of the developing regions where more than 90% of the population in these regions is pastoral. Besides, the federal and regional governments should give attention to local means of communication channels to promote the health of women and their children where most of the community has not access to television, radio, and other modern media channels. Additional systematic review and meta-analysis study is recommended to have a pooled estimation of a short birth interval and its determinants at the national level.

## Strengths and limitations of the study

This study was based on the most recent EDHS with a nationally representative large sample size. Moreover, this study applied multilevel modeling to handle the hierarchical nature of the EDHS data. Despite the above strengths, the study might have recall bias since the participants were asked about the events that took place 5 years or more preceding the survey. The study also shares the limitations of cross-sectional studies.

## Acknowledgments

The authors acknowledge the ICF International for Granting access to the use of the 2016 Ethiopian Demographic and Health Survey (EDHS) data for this study.

## Author Contributions

**Conceptualization:** Setognal Birara Aychiluhm, Kusse Urmale Mare, Mohammed Abdu.

**Data curation:** Setognal Birara Aychiluhm, Abay Woday Tadesse, Kusse Urmale Mare.

**Formal analysis:** Setognal Birara Aychiluhm, Abay Woday Tadesse.

**Methodology:** Setognal Birara Aychiluhm, Abay Woday Tadesse, Kusse Urmale Mare.

**Writing – original draft:** Setognal Birara Aychiluhm, Abay Woday Tadesse, Kusse Urmale Mare.

**Writing – review & editing:** Setognal Birara Aychiluhm, Abay Woday Tadesse, Kusse Urmale Mare, Mohammed Abdu, Abdusemed Ketema.

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
