## [Decision Letter · Decision Letter 0]

10 Jun 2020

PONE-D-20-07800

A Multilevel Analysis of Short Birth Interval and Its Determinants among Women in Pastoral Regions of Ethiopia

PLOS ONE

Dear Dr. Birara,

Thank you for submitting your manuscript to PLOS ONE. After careful consideration, we feel that it has merit but does not fully meet PLOS ONE’s publication criteria as it currently stands. Therefore, we invite you to submit a revised version of the manuscript that addresses the points raised during the review process.

We look forward to receiving your revised manuscript.

Kind regards,

Srinivas Goli, Ph.D.

Academic Editor

PLOS ONE

Journal Requirements:

2. We noticed you have some minor occurrence of overlapping text with previous publications:

https://doi.org/10.1186/s12905-019-0776-4

https://doi.org/10.1371/journal.pone.0226891

which needs to be addressed. In your revision ensure you cite all your sources (including your own works), and quote or rephrase any duplicated text outside the methods section. Further consideration is dependent on these concerns being addressed.

Please also discuss in your methods section, whether the data were de-identified before you accessed them

3. Please ensure that you refer to Figures 1 and 2 in your text as, if accepted, production will need this reference to link the reader to the figure.

Additional Editor Comments (if provided):

Dear Authors,

Review reports suggest that your manuscript has the merit. However, I agree with the reviewers suggestions. Considering the reviewers suggestions and my own reading of the paper, I suggest a major revision for this paper.

Regards

Srinivas

Reviewers' comments:

Reviewer's Responses to Questions

**Comments to the Author**

1. Is the manuscript technically sound, and do the data support the conclusions?

Reviewer #1: Partly

Reviewer #2: Partly

2. Has the statistical analysis been performed appropriately and rigorously? 

Reviewer #1: No

Reviewer #2: Yes

3. Have the authors made all data underlying the findings in their manuscript fully available?

Reviewer #1: Yes

Reviewer #2: Yes

4. Is the manuscript presented in an intelligible fashion and written in standard English?

Reviewer #1: No

Reviewer #2: No

5. Review Comments to the Author

Reviewer #1: Adequate birth spacing fetches great health benefits to both Mother and Child. The topic of the present study is of great interest for improving the Maternal and Child health in pastoral region of Ethiopia. The authors have used data from Demographic and Health Survey, 2016 on women in reproductive age-group 15-49 years. The present study aimed to address both the individual and community-related factors for short birth interval on four pastoral regions of Ethiopia using multilevel logistic regression technique. The manuscript does not contain line or page numbers which makes it difficult to point out observations. Therefore, I am mentioning the observations by sections in the manuscript.

Title

As the study is focused only on women in Child-bearing ages 15-49 years and not all women, the present title, "A Multilevel Analysis of Short Birth Interval and Its Determinants among Women in Pastoral Regions of Ethiopia" needs to be modified.

Abstract

" Avoid use of abbreviations in the abstract section.

" Please mention the four pastoral regions where the study was based.

" What is the sample size included in the study?

" Replace "independent predictor" with "predictor".

" The statement "The statistical significance level was declared at a 95% confidence interval" can be dropped from the abstract.

" The conclusion section needs to be strengthened, as the present one looks vague.

" The statement, "government should encourage local communication channels to promote the health of women and children" is ambiguous in interpretation. What measures are suggested by the authors to promote the health of women and children.

Introduction

" Paragraph 3 gives data related to Nigeria. Please provide similar data for Ethiopia.

" Also, total fertility rate is given per 1000 women. Please mention the units.

" In view of the statement "Like many other African countries, Ethiopia has shown so far little change in fertility reduction because of socio-cultural and religious factors." Can author give instances or examples of the cultural practices and religious factors responsible for higher fertility rates.

" Please mention the full form of the abbreviation SBI when first mentioned in the Introduction section.

" In view of the statement, "The Sustainable Development Goals (SDGs) of 2030, which combine multisystem strategies at global, regional and national levels, have three focuses to ensure healthy lives and promote wellbeing for all at all ages. Of these goals, one main objective is to reduce the neonatal mortality rate to lower than 12 per 1,000 live births [32-34] which is at a steady stage in developing nations." Can authors mention the present level of Infant and under five mortality in Ethiopia and Country specific goals to reduce IMR and U5MR (set by government or SDG).

" In view of the statement "Moreover, studies conducted in Ethiopia were limited and inconclusive to show the determinants of short birth interval at the community level (i.e. they were assessed only individual related factors). Therefore, this study aimed to address both the individual and community-related factors for short birth interval on four pastoral regions of Ethiopia using multilevel logistic regression analysis which is the appropriate model to handle community-level factors of short birth interval" it would be better to re-write the need for the study by adding more literature stating the importance of studying SBI in pastoral regions and use of multi-level analysis.

" It is not mentioned anywhere what is the meaning of pastoral region and why is it important to study it. It is difficult to understand the use of the present study for international audience.

" Dedicate the last paragraph of the Introduction section to mention only the need for the study and study objectives

Methods

Data Source

" What was the total sample collected by DHS in Ethiopia?

" What are these nine regions and two city administrations included by the survey?

" The statements "All women of reproductive-age group were included in the first stage" and "The information was collected from a nationally representative sample of 16,515 women aged 15-49 years" can be combined.

" Is child birth outside of wed-lock is common in Ethiopia? If not, unmarried women sample needs to be removed from the analysis. Were any measures taken to do the same?

" The author has used the term Pastoral region throughout the manuscript. There are a few questions which arise in my mind: 1) What is a pastoral region? , 2) What four pastoral regions have been selected in the study and what are the reasons for specifically selecting these regions? 3) What are the sample sizes from each of the regions? It would be better if the authors can provide an elaborate description on the sample selection.

" How have the authors adjusted for "twins" birth?

Ethics Statement: Although the present study rests on a publically available dataset, still as the study included human subjects, it is advised to add a section of ethical consideration in the methods section after data source.

Study Variables-Dependent Variable

" Use either of the two terms: Short Birth Interval or Optimal Birth Interval.

" The authors have calculated the birth interval as the time that elapsed between the birth date of the first child and the birth date of the second child which I think is the standard procedure. It is however advisable to include a reference of the same. You can find a reference here: https://dhsprogram.com/pubs/pdf/CR28/CR28.pdf (section 1.2).

" I would recommend using "Croft, T. N., Marshall, A. M., Allen, C. K., Arnold, F., Assaf, S., & Balian, S. (2018). Guide to DHS statistics. Rockville: ICF" to check and revise the computation procedure of the variables included in the study.

Independent Variables

" I am concerned regarding the choice of predictor variables. The predictor variables can be clubbed into following sections, namely reproductive, behavioral, and child status. However, it is not clear that which conceptual framework has been utilized to select these predictor variables. Please mention the same.

" Also, the abstract suggests that a multilevel multivariable analysis has been used, it is not clear what are the levels included for the analysis and which variable was introduced at which level? Please provide a detailed description of the same.

" The explanatory variables included in the study can be correlated with each other. Are any measures taken to check for the same? If yes, include a section explaining the same in the methods section after variable description.

Analysis Plan

" Data Analysis section needs to be restructured as there are a lot of repetition.

" The section "In data with a nested structure……. mixed- effect logistic regression analysis was used in this study." needs to be re-written as the meaning is not clear.

" The statement "Data were weighted before analysis and merge and re-categorize to make suitable for analysis" looks ambiguous. Please elaborate.

" A section on Bivariate analysis needs to be added before proceeding to the multivariate analysis. It is evident that Table 4 provides a 2x2 contingency table for all the predictor variables, however, it would be great if you can include chi-square/unadjusted odds ratio, and p-values in the table.

" Authors state that they have utilized software R, it is not clear which part of the analysis or data visualization was done in R.

Results

" Table 1 and 2 provides the descriptive account of the variables included in the study. These two tables can be combined to form a single table using sub-headings in the table.

" In both Table 1 and 2 the authors have mentioned Weighted frequency and percentage (unweighted) which needs to be substituted with unweighted frequency and weighted percentages.

" Include a row "Total" for the Table 1 and 2 or mention total sample size (N) in the table.

" The categorization of the explanatory variables needs an urgent attention, for instance variables like Religion, Respondent's educational Status, Husband's Educational Status, Respondent Occupation, Husbands Education have very small frequencies for certain categories which needs to be corrected.

" Additionally, are there any specific reason for categorizing number of births, number of living children, and Survival of index child as they are presently. Can these variables be used as continuous?

" For Media Exposure the variable is not directly available in the DHS dataset. What is computation procedure of the variable "Media Exposure". Also, what is the meaning of the categories "no" and "yes". Does "Yes" includes partial exposure to media? Please add a section describing the computation and categorization procedure in methods.

" Variables like ethnicity and social segregation play a vital role in affecting the behaviors and decision related to child birth and spacing. Are these variables present in the dataset? If yes, why are these not included in the analysis.

" Prevalence is generally not reported in Percentage, it is therefore recommended that the authors report prevalence per 100 individual (denominator). Also, it is advised to mention the exact prevalence (count) in the figure.

" Figure 1 seems unnecessary. It can be deleted.

" Add the data source and year to the headings of all the Tables and Figures.

" The titles of all the Tables and Figures needs to be modified more meaningfully.

" In the section "Prevalence of Short Birth Interval" the statement, "From a total of 2111(weighted) women who had at least two consecutive live births in four pastoral regions of Ethiopia", the authors have mentioned 2111 as weighted women, which needs to be substituted for unweighted number of women.

" As the existing literature points out that birth spacing can vary across the reproductive age-group. It is therefore advised to use the age-adjusted prevalence rates in the analysis.

Discussion

" The statement "This finding is higher than ……and United States (35%)". As the population size, socio-economic and developmental levels of Ethiopia and United States is quite different, it is better not to compare the two in discussion section.

" Also, the word "This finding" is an unclear reference. Please modify as required.

" The statement "Thus, the prevalence of SBI is …..and non- pastoral regions of the country" looks repetitive.

" In reference to the statement, "This could be…….In addition, this variation could be explained by a difference in cut-off values used to determine SBI". What were the definitions of SBI previously used (cut-offs).

Limitations of the Study

Please add a section on the limitations of the study.

Conclusions

" The statement, "government should encourage local communication channels to promote the health of women and children" is ambiguous in interpretation. What measures are suggested by the authors to promote the health of women and children.

" What is Xaagu system? How will it improve the present scenario?

Style of tabulation and presentation: The manuscript will benefit with major changes in the style of presentation. I would recommend the following paper published in PloS One (which I personally find extremely structured), to help the authors improve the style of tabulation and presentation:

1. Goli, S., Moradhvaj, A. R., & Shruti, J. P. (2016). High spending on maternity care in India: What are the factors explaining it?. PloS one, 11(6).

2. McNay, K., Arokiasamy, P., & Cassen, R. (2003). Why are uneducated women in India using contraception? A multilevel analysis. Population studies, 57(1), 21-40.

Language Issues

It is quite understandable that the authors of the manuscript are not native English speakers. There are grammatical and English language errors throughout the manuscript. The manuscript can be benefitted from an extensive grammar and language check. It is advised to take help from a native English speaker in order to achieve the English standard of the article published in PloS One.

Plagiarism

Thirteen percent of the text matches 15 sources or archives of academic publications. It is, therefore, advised to change the wording of Introduction and Methods sections. In majority of the instances the references are provided, however, the wordings of the entire paragraph are similar in a couple of occasions, which needs to be revisited and changed. I am unable to mention the exact paragraph which needs revision as no line numbers are provided in the manuscript. Majority of the text is similar to the following articles and report:

1. Birhanu, B. E., Kebede, D. L., Kahsay, A. B., & Belachew, A. B. (2019). Predictors of teenage pregnancy in Ethiopia: a multilevel analysis. BMC public health, 19(1), 601.

2. https://dhsprogram.com/data/Guide-to-DHS-Statistics/Place_of_Delivery.htm

3. Kawo, K. N., Asfaw, Z. G., & Yohannes, N. (2018). Multilevel analysis of determinants of anemia prevalence among children aged 6-59 Months in Ethiopia: classical and bayesian approaches. Anemia, 2018.

4. Woday, A., & Ayesheshim Muluneh, C. S. D. (2019). Birth asphyxia and its associated factors among newborns in public hospital, northeast Amhara, Ethiopia. PloS one, 14(12).

Reviewer #2: Short Birth Interval is a critical determinant of both maternal and child health and is an issue of concern in the developing world. The topic of the paper is an interesting one. However, here are some review points that might help improve the present work.

1. Authors can leave out the data collection method of DHS from the abstract. More importance should be given in explaining the tools and techniques used in the present research paper.

2. Introduction lacks continuity and flow. First authors should address why SBI is an important issue, the global scenario and then discuss its pertinence to African countries and the study region. Discussion of explanatory variables either should be better placed or put in the methods part where explanatory variables are listed out.

3. The need for a community-level study in the pastoral regions should be highlighted.

4. Why have the authors given weighted frequencies? It is difficult to understand the actual sample number that was collected. Only weighted percentage estimates should be enough.

5. Table 1 and 2 are both of explanatory variables. They can be merged with some partition within the table.

6. Rather than elaborating the explanatory variables more emphasis should be given on prevalence of SBI.

7. Figure 1, 2 are not of publishable quality.

8. Results on multilevel regression needs to be rewritten narrowing it down to only the results the authors find pertinent to the objectives of the study.

9. Usually wealth index of the household, women’s education play a significant role in determining spacing and limiting decisions, why are these variables not significant in Table 4? Authors might want to add it in the discussion.

10. No ante-natal care variables are taken in the study. During ante-natal care, women are exposed to various materials on how to practice spacing and limiting for the next birth. The multilevel analysis has not controlled for this variable.

11. Sex of preceding birth might be a better explanatory variable for SBI, as many communities around the globe has a preference for male child. If the preceding birth was female, there might be a shorter birth interval for the next child.

12. Authors should revisit the variables they have taken for multilevel analysis based on multicollinearity. Eg: No. of live births and No. of living children capture similar aspects of reproductive choices. It will be advisable to generate a cumulative score that captures all these measures together or choose the more critical one for the analysis.

13. Breastfeeding duration variable needs further explanation to understand whether this is for the preceding birth or all births. Breastfeeding duration for the index birth might not explain SBI for the last 2 births.

14. Authors highlight all previous studies were individual level and their study considers community-level factors. It will be more effective if a few community level variables are taken in the multilevel analysis, such as health infrastructural support, sanitation and hygiene practices in the neighbourhood, etc.

15. In discussion, authors merely summarise their results and point them to be similar to other studies. Discussion should have more content on implications of their results and suggest some policy revisions based on the findings.

16. Tense of the manuscript needs critical revision. Grammatical errors need to be reviewed. English editing might be beneficial.

All the best!

6. PLOS authors have the option to publish the peer review history of their article (what does this mean?). If published, this will include your full peer review and any attached files.

Reviewer #1: No

Reviewer #2: No

---

## [Author Response · Author response to Decision Letter 0]

25 Jul 2020

Dear Editors and reviewers, 

Thank you for giving us the opportunity to revise our manuscript entitled “A Multilevel Analysis of Short Birth Interval and Its Determinants among Reproductive age Women in Developing Regions of Ethiopia” before decisions. The authors have intensively discussed and addressed the raised concerns of the editor, and reviewers using point-by-point response as stated below. The amendments made on the manuscript have been presented using track change in the second attachment titled “Revised Manuscript with Track Changes”

Point-by-point responses for the questions and suggestions raised by Reviewer #1

Adequate birth spacing fetches great health benefits to both Mother and Child. The topic of the present study is of great interest for improving the Maternal and Child health in pastoral region of Ethiopia. The authors have used data from Demographic and Health Survey, 2016 on women in reproductive age-group 15-49 years. The present study aimed to address both the individual and community-related factors for short birth interval on four pastoral regions of Ethiopia using multilevel logistic regression technique. The manuscript does not contain line or page numbers which makes it difficult to point out observations. 

Therefore, I am mentioning the observations by sections in the manuscript.

Title

Q1. As the study is focused only on women in Child-bearing ages 15-49 years and not all women, the present title, "A Multilevel Analysis of Short Birth Interval and Its Determinants among Women in Pastoral Regions of Ethiopia" needs to be modified.

Response: We have revised it per raised constructive concerns of the reviewer.

Abstract

Q2. Avoid use of abbreviations in the abstract section.

Response: We have corrected it in the revised manuscript.

Q3." Please mention the four pastoral regions where the study was based.

Response: We have mentioned the four regions in the revised manuscript 

Q4" What is the sample size included in the study?

Response: 2683 total sample size (388 from Afar, 1706 from Somali, 489 from Benishangul-Gumuz, and 100 from Gambella )

Q5 " Replace "independent predictor" with "predictor".

Response: We have replaced it.

Q6 " The statement "The statistical significance level was declared at a 95% confidence interval" can be dropped from the abstract.

Response: We have dropped as suggested by the reviewer.

Q7" The conclusion section needs to be strengthened, as the present one looks vague.

Response: We have revised it based on your important suggestion. 

Q8" The statement, "government should encourage local communication channels to promote the health of women and children" is ambiguous in interpretation. What measures are suggested by the authors to promote the health of women and children.

Response: We have revised and incorporated in the revised manuscript.

Introduction

Q9" Paragraph 3 gives data related to Nigeria. Please provide similar data for Ethiopia.

Response: We have revised and incorporated data related to Ethiopia in the revised manuscript.

Q10" Also, total fertility rate is given per 1000 women. Please mention the units.

Response: We have revised and mentioned the units 

Q11" In view of the statement "Like many other African countries, Ethiopia has shown so far little change in fertility reduction because of socio-cultural and religious factors." Can author give instances or examples of the cultural practices and religious factors responsible for higher fertility rates.

Response: We have addressed it in the revised manuscript.

Q12" Please mention the full form of the abbreviation SBI when first mentioned in the Introduction section.

Response: We have corrected it.

Q13" In view of the statement, "The Sustainable Development Goals (SDGs) of 2030, which combine multisystem strategies at global, regional and national levels, have three focuses to ensure healthy lives and promote wellbeing for all at all ages. Of these goals, one main objective is to reduce the neonatal mortality rate to lower than 12 per 1,000 live births [32-34] which is at a steady stage in developing nations." Can authors mention the present level of Infant and under five mortality in Ethiopia and Country specific goals to reduce IMR and U5MR (set by government or SDG).

Response: We have incorporated the required information based on your important suggestion.

Q14" In view of the statement "Moreover, studies conducted in Ethiopia were limited and inconclusive to show the determinants of short birth interval at the community level (i.e. they were assessed only individual related factors). Therefore, this study aimed to address both the individual and community-related factors for short birth interval on four pastoral regions of Ethiopia using multilevel logistic regression analysis which is the appropriate model to handle community-level factors of short birth interval" it would be better to re-write the need for the study by adding more literature stating the importance of studying SBI in pastoral regions and use of multi-level analysis.

Response: We have revised it in the revised manuscript per raised constructive concerns of the reviewer.

Q15" It is not mentioned anywhere what is the meaning of pastoral region and why is it important to study it. It is difficult to understand the use of the present study for international audience.

Response: We have revised this section in revised manuscript 

Q16" Dedicate the last paragraph of the Introduction section to mention only the need for the study and study objectives

Response: We have revised this part as per the recommendation of the reviewer.

Methods

Data Source

Q17" What was the total sample collected by DHS in Ethiopia?

Response: a total sample size of 16,515 women aged 15–49 years was collected. 

Q18" What are these nine regions and two city administrations included by the survey?

Response: We have corrected it in the revised manuscript. The nine regions are Tigray, Afar, Amhara, Oromia, Somali, Benishangul-Gumuz, Southern Nations Nationalities and Peoples Region (SNNPR), Gambella, and Harari), and the two city administrations are (Addis Ababa and Dire-Dawa). 

Q19" The statements "All women of reproductive-age group were included in the first stage" and the information was collected from a nationally representative sample of 16,515 women aged 15-49 years" can be combined.

Response: We have corrected it as per suggested by the reviewer. 

Q20" Is child birth outside of wed-lock is common in Ethiopia? If not, unmarried women sample needs to be removed from the analysis. Were any measures taken to do the same?

Response: We have revised it in the main document. Women who had never been married and those who have multiple births out of wedlock were excluded from the analysis.

" The author has used the term Pastoral region throughout the manuscript. There are a few questions which arise in my mind: 

Q21 1) What is a pastoral region? 

Response: we have revised this term and substituted pastoral region by developing region to make it clear for the audiences and we have briefly elaborated in the study area and data source section of the revised manuscript. 

Q22 2) What four pastoral regions have been selected in the study and what are the reasons for specifically selecting these regions? 

Response: We have revised this concern and incorporated all the required information in the revised manuscript.

Q23 3) What are the sample sizes from each of the regions? It would be better if the authors can provide an elaborate description on the sample selection.

Response: the authors corrected and incorporated the required data in the revised version of the manuscript. The sample sizes from each of the regions were 388 from Afar, 1706 from Somali, 489 from Benishangul-Gumuz, and 100 from Gambella

 Q24 " How have the authors adjusted for "twins" birth?

Response: We have described in the revised version of the manuscript under study area and data source section. 

Q25 Ethics Statement: Although the present study rests on a publically available dataset, still as the study included human subjects, it is advised to add a section of ethical consideration in the methods section after data source.

Response: We have revised this concern and included ethics statement in the main revised manuscript. 

Study Variables-Dependent Variable

Q26" Use either of the two terms: Short Birth Interval or Optimal Birth Interval.

Response: We have corrected it in the revised manuscript

Q27" The authors have calculated the birth interval as the time that elapsed between the birth date of the first child and the birth date of the second child which I think is the standard procedure. It is however advisable to include a reference of the same. You can find a reference here: https://dhsprogram.com/pubs/pdf/CR28/CR28.pdf(section 1.2).

Response: we have cited the recommended reference in the revised version of the manuscript

Q28" I would recommend using "Croft, T. N., Marshall, A. M., Allen, C. K., Arnold, F., Assaf, S., & Balian, S. (2018). Guide to DHS statistics. Rockville: ICF" to check and revise the computation procedure of the variables included in the study.

Response: Thank you very much for your recommendation, we have considered this guideline for computation of this DHS dataset.

Independent Variables

Q29" I am concerned regarding the choice of predictor variables. The predictor variables can be clubbed into following sections, namely reproductive, behavioral, and child status. However, it is not clear that which conceptual framework has been utilized to select these predictor variables. Please mention the same.

Response: this concern is incorporated in the revised version of the manuscript under study variable section. we have developed the conceptual framework by reviewed literature published before. All the independent variables were selected based on those literature.

Q30" Also, the abstract suggests that a multilevel multivariable analysis has been used, it is not clear what are the levels included for the analysis and which variable was introduced at which level? Please provide a detailed description of the same.

Response: we have revised this issue and incorporated in the revised version of the manuscript under the independent variable section. Here in our study, independent variables were classified in to individual level variables and community level variables.

Q31" The explanatory variables included in the study can be correlated with each other. Are any measures taken to check for the same? If yes, include a section explaining the same in the methods section after variable description.

Response: we have corrected and included a section which describes this concern in the revised version of the manuscript.

Analysis Plan

Q32" Data Analysis section needs to be restructured as there are a lot of repetition.

Response: we have corrected this concern in the revised version of the manuscript 

Q33" The section "In data with a nested structure……. mixed- effect logistic regression analysis was used in this study." needs to be re-written as the meaning is not clear.

Response: we have corrected this concern in the revised version of the manuscript 

Q34" The statement "Data were weighted before analysis and merge and re-categorize to make suitable for analysis" looks ambiguous. Please elaborate.

Response: this concern is revised and incorporated in the revised version of the manuscript.

Q35" A section on Bivariate analysis needs to be added before proceeding to the multivariate analysis. It is evident that Table 4 provides a 2x2 contingency table for all the predictor variables, however, it would be great if you can include chi square/unadjusted odds ratio, and p-values in the table.

Response: we have addressed in the revised version of the manuscript. In this contingency table the frequency of each categories of predictors is putted to compute crude/unadjusted odds ratio, thus, including the crude/unadjusted odds ratio in this table will be redundancy. As we know all, to know statistically significant predictors in multivariable analysis, we can use either p value (i.e if the value less than 0.05) or confidence interval of adjusted odds ratio (i.e if the confidence interval not include unity or 1). Here in our case, we putted the AOR with its corresponding confidence interval to know statistically significant predictors. So, we believed that including the p value will not add any new information if the AOR with its confidence interval is given.

Q36" Authors state that they have utilized software R, it is not clear which part of the analysis or data visualization was done in R.

Response: We have used R for graphical presentation of data (i.e. Fig 1 and Fig 2), we have corrected this concern in the revised version of the manuscript.

Results

Q37" Table 1 and 2 provides the descriptive account of the variables included in the study. These two tables can be combined to form a single table using sub-headings in the table.

Response: we have revised it as suggested.

Q38" In both Table 1 and 2 the authors have mentioned Weighted frequency and percentage (unweighted) which needs to be substituted with unweighted frequency and weighted percentages.

Response: we have revised it as suggested. Based on the recommendation of the EDHS 2016 report, we have used sample weight for the frequency as well as for the percentage. 

Q39" Include a row "Total" for the Table 1 and 2 or mention total sample size (N) in the table.

Response: we have corrected as suggested and we include total sample size(N) in the table.

Q40" The categorization of the explanatory variables needs an urgent attention, for instance variables like Religion, Respondent's educational Status, Husband's Educational Status, Respondent Occupation, Husbands Education have very small frequencies for certain categories which needs to be corrected.

Response: we have revised and recategorized those independent variables as suggested.

Q41" Additionally, are there any specific reason for categorizing number of births, number of living children, and Survival of index child as they are presently. Can these variables be used as continuous?

Response: we have revised this concern in the revised manuscript. We have used their continuous form for predictors number of births and number of living children based on your important suggestion but the predictor survival of index child is categorical by its nature ( i.e survived or not) in the EDHS data set, So we have used as it is.

Q42" For Media Exposure the variable is not directly available in the DHS dataset. What is computation procedure of the variable "Media Exposure ". 

Also, what is the meaning of the categories "no" and "yes". Does "Yes" includes partial exposure to media? Please add a section describing the computation and categorization 

procedure in methods.

Response: this concern is revised and incorporated in the revised version of the manuscript.

Q43" Variables like ethnicity and social segregation play a vital role in affecting the behaviors and decision related to child birth and spacing. Are these variables present in the dataset? If yes, why are these not included in the analysis.

Response: Social segregation was not present in the dataset; ethnicity is available in the dataset however there were no variability in respondents to the predictor ethnicity. Due to this we did not included it in the analysis.

Q44" Prevalence is generally not reported in Percentage, it is therefore recommended that the authors report prevalence per 100 individual (denominator). Also, it is advised to mention the exact prevalence (count) in the figure.

Response: The authors addressed this concern in the revised manuscript.

Q45" Figure 1 seems unnecessary. It can be deleted.

Response: we have deleted it as suggested 

Q46" Add the data source and year to the headings of all the Tables and Figures.

Response: we have included the data source (i.e EDHS) and year (i.e 2016). 

Q47" The titles of all the Tables and Figures needs to be modified more meaningfully.

Response: we have revised it as suggested. 

Q48" In the section "Prevalence of Short Birth Interval" the statement, "From a total of 2111(weighted) women who had at least two consecutive live births in four pastoral regions of Ethiopia", the authors have mentioned 2111 as weighted women, which needs to be substituted for unweighted number of women.

Response: As we know, the allocation of the sample in the EDHS data to different regions as well as urban and rural areas were non-proportional, therefore based on the recommendation of EDHS 2016 report, all proportions and frequencies were estimated after applying sample weights to the data to adjust for disproportionate sampling and non-responses. We have explained the detail on data analysis section of the revised manuscript. Based on your suggestion we have included the unweighted number of women in the commented statement under “Prevalence of Short Birth Interval section” of the revised manuscript.

Q49" As the existing literature points out that birth spacing can vary across the reproductive age-group. It is therefore advised to use the age-adjusted prevalence rates in the analysis.

Response: we have revised it as suggested by the reviewer. 

Discussion

Q50" The statement "This finding is higher than ……and United States (35%)". As the population size, socio-economic and developmental levels of Ethiopia and United States is quite different, it is better not to compare the two in discussion section.

Response: we have corrected it as suggested.

Q51" The statement "Thus, the prevalence of SBI is …..and non- pastoral regions of the country" looks repetitive.

Response: we have removed the repetition as suggested.

Q52" In reference to the statement, "This could be…….In addition, this variation could be explained by a difference in cut-off values used to determine SBI". What were the definitions of SBI previously used (cut-offs).

Response: Those previous studies considered SBI, if birth interval less than 36 months, whereas our study defined it as less than 24 months.

Q53 Limitations of the Study

Response: we have included limitation of study in the revised manuscript.

Conclusions

Q54" The statement, "government should encourage local communication channels to promote the health of women and children" is ambiguous in interpretation. What measures are suggested by the authors to promote the health of women and children.

Response: we have revised this section in the revised manuscript. The government should mobilize the community using health extension workers, women’s group like Women’s development Army (WDA). 

Q55" What is Xaagu system? How will it improve the present scenario?

Response: we have revised this section in the revised manuscript. Xaagu or dagu system is a local means for news exchange. It is a social institution with particular purposes in the daily life of the Afar communities. It functions within a defined set of regulations and expectations, though the rules are necessarily unwritten. The law of dagu means that whenever you meet someone on the road who has travelled reasonably far, say from a nearby village, you are required to pause and

engage in a news exchange session.

Q56 Style of tabulation and presentation: The manuscript will benefit with major changes in the style of presentation. I would recommend the following paper published in PloS One (which I personally find extremely structured), to help the authors improve the style of tabulation and presentation:

1. Goli, S., Moradhvaj, A. R., & Shruti, J. P. (2016). High spending on maternity care in India: What are the factors explaining it?. PloS one, 11(6).

2. McNay, K., Arokiasamy, P., & Cassen, R. (2003). Why are uneducated women in India using contraception? A multilevel analysis. Population studies, 57(1), 21-40.

Response: Thank you very much for your recommendation, we have seen those recommended articles and revised the tabulation and presentation style of our manuscript in the revised version of the manuscript.

Language Issues

Q57 It is quite understandable that the authors of the manuscript are not native English speakers. There are grammatical and English language errors throughout the manuscript. The manuscript can be benefitted from an extensive grammar and language check. It is advised to take help from a native English speaker in order to achieve the English standard of the article published in PloS One.

Response: Thank you for your important suggestion. We have addressed in the revised version of the manuscript.

Plagiarism

Q58 Thirteen percent of the text matches 15 sources or archives of academic publications. It is, therefore, advised to change the wording of Introduction and Methods sections. In majority of the instances the references are provided, however, the wordings of the entire paragraph are similar in a couple of occasions, which needs to be revisited and changed. I am unable to mention the exact paragraph which needs revision as no line numbers are provided in the manuscript. Majority of the text is similar to the following articles and report:

1. Birhanu, B. E., Kebede, D. L., Kahsay, A. B., & Belachew, A. B. (2019). Predictors of teenage pregnancy in Ethiopia: a multilevel analysis. BMC public health, 19(1), 601.

2. https://dhsprogram.com/data/Guide-to-DHS-Statistics/Place_of_Delivery.htm

3. Kawo, K. N., Asfaw, Z. G., & Yohannes, N. (2018). Multilevel analysis of determinants of anemia prevalence among children aged 6-59 Months in Ethiopia: classical and bayesian approaches. Anemia, 2018.

4. Woday, A., & Ayesheshim Muluneh, C. S. D. (2019). Birth asphyxia and its associated factors among newborns in public hospital, northeast Amhara, Ethiopia. PloS one, 14(12).

Response: Thank you for your crucial advice. The authors addressed this issue in the revised version of the manuscript.

Point-by-point responses for the questions and suggestions raised by Reviewer #2

Short Birth Interval is a critical determinant of both maternal and child health and is an issue of concern in the developing world. The topic of the paper is an interesting one. However, here are some review points that might help improve the present work.

1. Authors can leave out the data collection method of DHS from the abstract. More importance should be given in explaining the tools and techniques used in the present research paper.

Response: we have revised it as suggested 

2. Introduction lacks continuity and flow. First authors should address why SBI is an important issue, the global scenario and then discuss its pertinence to African countries and the study region. Discussion of explanatory variables either should be better placed or put in the methods part where explanatory variables are listed out.

Response: we have revised it as suggested

3. The need for a community-level study in the pastoral regions should be highlighted.

Response: we have addressed this issue in the revised manuscript.

4. Why have the authors given weighted frequencies? It is difficult to understand the actual sample number that was collected. Only weighted percentage estimates should be enough.

Response: Since, the allocation of the sample in the EDHS to different regions as well as urban and rural areas were non-proportional, therefore based on the recommendation of EDHS 2016 report, all proportions and frequencies were estimated after applying sample weights to the data to adjust for disproportionate sampling and non-responses. We have explained the detail on data analysis section of the revised manuscript.

5. Table 1 and 2 are both of explanatory variables. They can be merged with some partition within the table.

Response: we have merged it as suggested 

6. Rather than elaborating the explanatory variables more emphasis should be given on prevalence of SBI.

Response: the authors have addressed this concern in the revised manuscript 

7. Figure 1, 2 are not of publishable quality.

Response: we have corrected it based on the suggestions of you and reviewer #1 in the revised manuscript

8. Results on multilevel regression needs to be rewritten narrowing it down to only the results the authors find pertinent to the objectives of the study.

Response: we have tried to address it based on our objectives

9. Usually wealth index of the household, women’s education plays a significant role in determining spacing and limiting decisions, why are these variables not significant in Table 4? Authors might want to add it in the discussion.

Response: the authors addressed this concern. As you said that wealth index of the household, women’s education predictors of birth interval, in our study women’s education (category secondary education and above level) is significantly associated with short birth interval. But wealth index of the household not statistically significant. 

10. No ante-natal care variables are taken in the study. During ante-natal care, women are exposed to various materials on how to practice spacing and limiting for the next birth. The multilevel analysis has not controlled for this variable.

Response: You are wright but this variable is not available in this data set.

11. Sex of preceding birth might be a better explanatory variable for SBI, as many communities around the globe has a preference for male child. If the preceding birth was female, there might be a shorter birth interval for the next child.

Response: Thank you very much for your important concern. In our case sex of preceding birth is one of the important predictors of short birth interval that is women having female sex of the index child had 1.13 times greater risk of short birth interval compared to those women having male index children.

12. Authors should revisit the variables they have taken for multilevel analysis based on multicollinearity. Eg: No. of live births and No. of living children capture similar aspects of reproductive choices. It will be advisable to generate a cumulative score that captures all these measures together or choose the more critical one for the analysis.

Response: we have considered the issue of multicollinearity in multivariable multilevel analysis and we have included a section described about this concern in the revised version of the manuscript.

13. Breastfeeding duration variable needs further explanation to understand whether this is for the preceding birth or all births. Breastfeeding duration for the index birth might not explain SBI for the last 2 births.

Response: Breastfeeding duration is asked for the preceding index child and it is known that women who breastfeed their children for longer duration have a longer period of amenorrhea which results in postpartum infertility. So that it determines the birth interval of the next births of a woman.

14. Authors highlight all previous studies were individual level and their study considers community-level factors. It will be more effective if a few community level variables are taken in the multilevel analysis, such as health infrastructural support, sanitation and hygiene practices in the neighbourhood, etc.

Response: Thank you for your important suggestion, Variables like health infrastructural support, sanitation and hygiene practices in the neighbourhood not available in the EDHS dataset. However, variables like residence site, region, and cluster are available in the dataset. The authors included those available variables in the multilevel analysis. The detail is explained in the variables section of the revised manuscript.

15. In discussion, authors merely summarise their results and point them to be similar to other studies. Discussion should have more content on implications of their results and suggest some policy revisions based on the findings.

Response: The authors tried to review this concern in the revised version of the manuscript 

16. Tense of the manuscript needs critical revision. Grammatical errors need to be reviewed. English editing might be beneficial.

Response: Thank you for your important advice. The authors addressed this issue in the revised version of the manuscript.

---

## [Editor Report · Decision Letter 1]

30 Jul 2020

A Multilevel Analysis of Short Birth Interval and Its Determinants among Reproductive age Women in Developing Regions of Ethiopia

PONE-D-20-07800R1

Dear Dr. Birara,

We’re pleased to inform you that your manuscript has been judged scientifically suitable for publication and will be formally accepted for publication once it meets all outstanding technical requirements.

Kind regards,

Srinivas Goli, Ph.D.

Academic Editor

PLOS ONE

Additional Editor Comments (optional):

The revisions are satisfactory. 
---

## [Editor Report · Acceptance letter]

14 Aug 2020

PONE-D-20-07800R1 

A Multilevel Analysis of Short Birth Interval and Its Determinants among Reproductive age Women in Developing Regions of Ethiopia 

Dear Dr. Aychiluhm:

I'm pleased to inform you that your manuscript has been deemed suitable for publication in PLOS ONE. Congratulations! Your manuscript is now with our production department. 

Kind regards, 

on behalf of

Dr. Srinivas Goli 

Academic Editor

PLOS ONE